# Breeding for Virus Resistance and Its Effects on Deformed Wing Virus Infection Patterns in Honey Bee Queens

**DOI:** 10.3390/v13061074

**Published:** 2021-06-04

**Authors:** David Claeys Bouuaert, Lina De Smet, Dirk C. de Graaf

**Affiliations:** Laboratory of Molecular Entomology and Bee Pathology, Ghent University, Krijgslaan 281, B-9000 Ghent, Belgium; lina.desmet@ugent.be (L.D.S.); Dirk.deGraaf@UGent.be (D.C.d.G.)

**Keywords:** honey bee, suppressed in ovo virus infection, virus resistance, deformed wing virus

## Abstract

Viruses, and in particular the deformed wing virus (DWV), are considered as one of the main antagonists of honey bee health. The ‘suppressed in ovo virus infection’ trait (SOV) described for the first time that control of a virus infection can be achieved from genetically inherited traits and that the virus state of the eggs is indicative for this. This research aims to explore the effect of the SOV trait on DWV infections in queens descending from both SOV-positive (QDS+) and SOV-negative (QDS–) queens. Twenty QDS+ and QDS– were reared from each time four queens in the same starter–finisher colony. From each queen the head, thorax, ovaries, spermatheca, guts and eviscerated abdomen were dissected and screened for the presence of the DWV-A and DWV-B genotype using qRT-PCR. Queens descending from SOV-positive queens showed significant lower infection loads for DWV-A and DWV-B as well as a lower number of infected tissues for DWV-A. Surprisingly, differences were less expressed in the reproductive tissues, the ovaries and spermatheca. These results confirm that selection on the SOV trait is associated with increased virus resistance across viral genotypes and that this selection drives DWV towards an increased tissue specificity for the reproductive tissues. Further research is needed to explore the mechanisms underlying the interaction between the antiviral response and DWV.

## 1. Introduction

Over the last decades, honey bee health has been severely affected by the presence of the ectoparasitic mite *Varroa destructor* [1,2,3,4,5]. As the mite feeds on the fat body of the honey bee [6], virus transmission can occur [7]. During this process, the main organ evolved to provide protection to environmental pathogens—the gut—is bypassed [8]. This new transmission pathway has led to an increase in virulence and genotype diversity of the deformed wing virus (DWV) [9,10,11]. A process that is enhanced by beekeeping management practices aimed at minimizing disease prevalence [12]. Infection with DWV can either result in low-level infections with asymptomatic bees or high-level infections resulting in clinical symptoms such as deformed wings, shortened abdomens, discolorations, behavior abnormalities and reduced lifespan [13,14]. Notwithstanding the lack of clinical symptoms, low level infections—often combined with other stressors—have been linked with increased colony losses [13,14,15,16,17,18], especially during the winter [2,19]. Being reported in 90% of all apiaries [15,16] and over all honey bee castes and developmental stages [13], DWV can be considered as one of the most important threats for honey bee health.

The name ‘deformed wing virus’ groups multiple genetically similar viruses of the Iflaviridae family [13]. At present, DWV is commonly subdivided in the genotypes DWV-A and DWV-B, although recombination between the two genotypes is common [17]. DWV-A is considered to be more closely related to the isolate of the DWV as it was first described [17], whereas DWV-B (formally *Varroa destructor* virus-1 [18]) is shown to replicate in varroa mites and is assumed to be diverged from DWV-A following the arrival of the varroa mite [7,9,19]. The subdivision of DWV in genotypes represent different consensus sequences. Members of the Iflaviridae have limited template-copying fidelity and lack proofreading and repair functions. The high number of mutants formed by the error-prone replication process generate an ever-changing mutant cloud described as a quasispecies. Virus particles of the same genotype are thus never identical but share a high similarity with the consensus sequence defined by the genotype [20].

Transmission of viruses within and between colonies can be divided in vertical and horizontal transmission. Horizontal transmission is the spread of a virus between individuals of the same generation through either bodily contact, trophallaxis, feeding or common flower visits. Vertical transmission is the spread of a virus from parents to their offspring and can occur via eggs or semen [21]. Where horizontal transmission has an important link with varroa infestations, vertical transmission is presumably important for the maintenance of DWV in honey bee populations [22]. As the sole responsible for egg laying, honey bee queens play an important role in vertical transmission. This is reflected by the high presence of DWV encountered in the reproductive tissues, ovaries, spermatheca, and eggs, [23,24,25,26,27] which may be an adaptation to enhance vertical transmission [22]. Transmission from the queen’s ovaries to her eggs occurs predominantly by virus particles adhering to the surface of the eggs (transovum) [22]. Despite being commonly found in the study by Amiri and colleagues [23], a quantitative assessment of the virus copy number in the eggs and ovaries suggested low transmission efficiencies and a high dependency of the infection level of the queen [22]. This complies with the findings that not all offspring from DWV infected queens are infected with DWV [21,28] and that the pathogen profile of queens does not reflect that of the workers [29].

In 2012, Belgian beekeepers started screening breeding queens for the presence of viruses in eggs deposited in worker brood cells and found 75% to be infected with at least one virus and 40% to be infected with DWV [26]. These high results led to the start of a yearly sanitary control of the breeding queens and eventually to the discovery of the ‘Suppressed in ovo virus infection’ trait (SOV) [27]. This trait is described by the absence of virus infections in 10 pooled drone eggs and was the first study to show that the potential of honey bees to suppress virus infections is heritable (R ≈ 0.25). In addition, colonies headed by a SOV-positive queen have fewer and less severe DWV infections in almost all developmental phases of both worker bees and drones [27] indicating the beneficial implications for the health status of the colony. The underlying mechanisms of the SOV trait are yet to be discovered; possible hypotheses could be an increase in the expression [30] or effectivity [17] of the RNA interference pathway, transgenerational effects [31] or increased expression levels of Toll-6 which has been linked with survival after artificial infection with Israeli acute paralysis virus (IAPV) [32].

So far, the SOV trait has not yet been described from the perspective of the queen and has been focused on the DWV-A genotype. In this paper, the infection patterns of DWV in the different tissues (head, thorax, spermatheca, ovaries, guts and eviscerated abdomen) of queens descending from SOV positive (QDS+) and SOV negative (QDS–) queens are presented. In addition, DWV is quantified for both the DWV-A and DWV-B genotype to gain insights in the effects of the increased virus resistance associated with the SOV trait on different genotypes. As viruses are one of the most important threats for honey bee health, a better understanding of the interaction between virus resistance and the virulence and transmission characteristics of viruses is of paramount importance to increase honey bee health globally.

## 2. Materials and Methods

### 2.1. Sample Collection

Four SOV-positive and four SOV-negative honey bee colonies were randomly selected from the breeding stock of Ghent University. All colonies were healthy, did not show signs of disease and were either free of viruses or infected with DWV only (determined during the spring drone laying season according to the protocol described by de Graaf and colleagues [27] as part of the ongoing SOV screening). Of each colony, five queens were reared following standard queen rearing practices [33]. All queens, descending from either SOV+ or SOV- breeding queens were transported to the same queenless starter–finisher colony after grafting. All colonies were grafted within a 3-week timespan. The starter–finisher colony was created from a strong colony, free of disease symptoms. After closure of the queen cups (5–6 days later) all queen cups were caged and transported to an incubator kept at 35 °C and 70% humidity. Each caged queen cup received a small amount of sugar candy to survive the first hours. Twice a day, queens were checked for emergence and dissected the same day.

### 2.2. Dissection

Queens were killed by freezing and dissected immediately following guidelines described by Carreck and colleagues [34]. Head, thorax, guts, ovaries, spermatheca and the eviscerated abdomen (cleared from guts, ovaries and spermatheca) were stored separately in 500 µL QIAzol lysis reagent (Qiagen, Hilden, Germany) at −20 °C until further analyzes on the viral status.

### 2.3. RNA Extraction and cDNA Synthesis

Samples were homogenized using a Tissuelyser II in the presence of zirconium beads. RNA was extracted using the RNeasy Lipid tissue mini kit (Qiagen, Hilden, Germany) according to the manufacturer’s instructions and eluted in 30 µL elution buffer. RNA was reverse-transcribed with the RevertAid H Minus First Strand cDNA Synthesis Kit (Thermo Scientific, Waltham, MA, United States) using random hexamer primers starting from 5 µL RNA.

### 2.4. qRT-PCR

Determining the DWV load in all samples was done by uniplex qRT-PCR using Platinum SYBR Green qPCR SuperMix-UDG (Thermo Scientific, Waltham, MA, United States). Each reaction mix consisted of 0.4 µM of each primer (see Appendix A), 11.45 µL RNase-free water, 12.5 µL SYBR Green and 1 µL of cDNA template. Samples were run in duplicate in a three-step real-time qPCR with following thermal cycling conditions: denaturation stage at 95 °C—15 s, annealing stage at 58 °C—20 s and extension stage at 72 °C—30 s for 35 cycles. This procedure was followed by a melt-curve dissociation analysis to confirm the specificity of the product (55–95 °C with an increment of 0.5 °C s^−1^). Each plate included a no template control (NTC). Absolute quantification was based on a standard curve obtained through an eightfold 5× serial dilution of viral plasmid control that ranged between 10^4^–10^10^ copies/µL. All data were analyzed using CFX Manager 3.1 software (Bio-Rad, Hercules, California, United States). Baseline correction and threshold setting were automatically calculated by the software. Maximum accepted quantification cycle (Ct) difference between replicates was set to one Ct. The successful amplification by RT-PCR of the reference gene β-actin following the protocol described by de Graaf and colleagues [27] was used to confirm the integrity of samples throughout the entire procedure.

### 2.5. Statistics

Viral loads for each sample were log10-transformed to improve data visualization. RStudio version 3.6.1 ( RStudio, Boston, MA, USA) was used for data analysis and visualization. Analyses of the differences in number of infections and infection load were conducted by chi-squared tests, Wilcoxon Rank Sum tests or Student’s *t*-Tests depending on the sample size and data type. All tests were checked for and complied with the required assumptions.

## 3. Results

### 3.1. Infections

All queens were infected in at least one tissue for both DWV-A and DWV-B. Figure 1 shows the distribution of the number of infected tissues in each individual queen for both DWV-A and DWV-B and for QDS– and QDS+. The average number of infected tissues is significantly lower for QDS+ (2.2) compared to the QDS– (4.0) for DWV-A (Independent sample *t*-test t(237.74) = 4.85, *p* < 0.01).

Significantly lower number of infections were found in QDS+ compared to QDS– for DWV-A in the head (chi-squared (1, N = 40) = 6.23, *p* < 0.05), thorax (chi-squared (1, N = 40) = 6.25, *p* < 0.05) and eviscerated abdomen (chi-squared (1, N = 40) = 6.25, *p* < 0.05; Figure 2). DWV-B was found in most guts, ovaries, spermatheca and eviscerated abdomen but only rarely in the thorax and head. No significant differences were found between QDS+ and QDS– in the number of samples infected with DWV-B for each tissue (Figure 2). No influence of the hive from where queens originated was found on the number of infections eliminating a disproportionate effect of heavily infected hives over other hives (Wilcoxon Rank Sum tests W = 27982, *p* = 0.87). Appendix A provides a summary table of the number of infections and the average infection load per tissue for DWV-A and DWV-B.

### 3.2. Average Infection Load

The average infection load for DWV-A was significantly lower in QDS+ compared to QDS– over all tissues (Independent sample *t*-test t(102.17) = 2.04, *p* < 0.05). In addition, infection loads were significantly lower in QDS+ compared to QDS– for DWV-A in the ovaries (Independent sample *t*-test t(22.69) = 2.85, *p* < 0.01), spermatheca (Independent sample *t*-test t(16.88) = 4.66, *p* < 0.01), guts (Independent sample *t*-test t(15.25)= 4.63, *p* < 0.01) and eviscerated abdomen (Independent sample *t*-test t(13.58)= 4.44, *p* < 0.01; Figure 3). Comparing the average infection load for DWV-B showed significant lower infection loads for QDS+ in the guts (Independent sample *t*-test t(25.93)= 2.95, *p* < 0.01) and eviscerated abdomen (Independent sample *t*-test t(29.98)= 3.89, *p* < 0.01; Figure 3). No significant differences were found between the average infection load of DWV-B between QDS+ and QDS– over all tissues combined. For the average infection load of the head and the thorax, insufficient tissues were infected for statistical analyses.

## 4. Discussion

All honey bee queens were commonly infected with both DWV-A and DWV-B in all tissues with the exception of DWV-B in the head and thorax. The significant reduction in the number of infected tissues and the significant lower infection load confirms that the SOV trait is associated with increased virus resistance and that this increased resistance is expressed in the descending honey bee queens. In contrast to what was expected, no significant differences were found in the number of infected reproductive tissues (ovaries and spermatheca) between QDS+ and QDS–, albeit being lower in QDS+ for DWV-A. As a response to the reduced vertical transmission via eggs expressed by the SOV trait the DWV quasispecies appears to be selected towards reduced virulence (lower infection loads) and increased tissue specificity for the reproductive tissues (lower effect size in the ovaries and spermatheca). These patterns confirm the role of vertical transmission as a key strategy for DWV-A [35] and the tissue specificity of DWV for the ovaries [21,23] and spermatheca [24]. In addition, both patterns comply with exiting theories on the effect of selection for increased virus resistance on viral quasispecies [16,36,37,38]. The bottleneck caused by the increased virus resistance could imply a reduction of the quasispecies diversity. This process has been shown to be associated with decreased viral fitness, virulence and pathogenesis as it limits the ability to overcome innate barriers within and between tissues [39,40,41,42]. On a cellular level, the interaction between the host antiviral response and the viral quasispecies can evolve separately within different tissues of the same host as has been shown for tissue specificity and transmission patterns of HIV over time [43,44,45]. A similar interaction has been shown in insects were RNAi-mediated responses drives quasispecies diversity affecting virulence and transmission patterns [46]. Thus far, interactions between honey bee antiviral responses and viral quasispecies diversity remains unknown. Further research is needed to conclude if the interaction between the increased virus resistance associated with the SOV trait and the viral quasispecies diversity acts as an underlying mechanism behind the SOV trait.

The screening protocol for the SOV trait initially focused on DWV-A [27]. With the inclusion of DWV-B in this study, the specificity of the increased virus resistance on different genotypes is described for the first time in honey bees queens. As shown by the results, differences in the average infection loads between QDS+ and QDS– are similar for DWV-A compared to DWV-B although the effects on the number of infected tissues is only significant for DWV-A. This confirms that selection for virus resistance is not restricted to a specific genotype, which aligns with the heritability of the SOV trait for both individual viruses and for the total number of virus infections [27]. The stronger effect of DWV-A compared to DWV-B might be attributed to the association between DWV-B and transmission via the varroa mite [17,19], reducing the importance of vertical transmission for the DWV-B genotype.

Possible infections occurring during larval feeding might be responsible for the high number of infected queens. Although larvae have a marked resistance to viral infection via oral transmission [17], transmission of viruses through larval feeding has been described [21,47]. This implies that the possible effect of the differences in the initial infection of the eggs between the SOV positive and SOV negative colonies cannot be isolated in this study. Repeating the experiments by in-vitro rearing the honey bee queens could distinguish between the effects of infections transmitted by the eggs and infections transmitted during larval feeding. As the experiment was stopped before the QDS+ and QDS– queens started oviposition, no information is available on the SOV status of the queens themselves. In order to investigate the impact of the infection patterns in queens on that of her offspring, further experiments need to be set up. This accounts as well for the effects of the queen’s age on the infection patterns as antiviral responses [48] and infection patterns [24,49,50] are thought to vary with age.

Thus far, only a limited number of studies have focused on breeding for virus resistance [27,32] despite the relevance of viruses for the health status of honey bee queens [51] and of the colony as a whole [52,53,54]. The results presented in this study confirm that breeding for virus resistance can achieve significant improvements over a limited number of generations. As all honey bee queens were infected with both genotypes of DWV, especially in the ovaries and spermatheca, it can be concluded that the SOV trait does not result in a clearing of the viral infection in the offspring queens, but rather in an downscaling of the infection level. This shows that the SOV trait can act as an important tool in the objective to strive towards a stable co-evolution between honey bees and their pathogens. As the SOV trait has a genetic background further research on the genetic markers associated with the trait could contribute to unraveling the underlying mechanisms of the trait and its implications on the interaction with the DWV quasispecies.

## Figures and Tables

**Figure 1 viruses-13-01074-f001:**
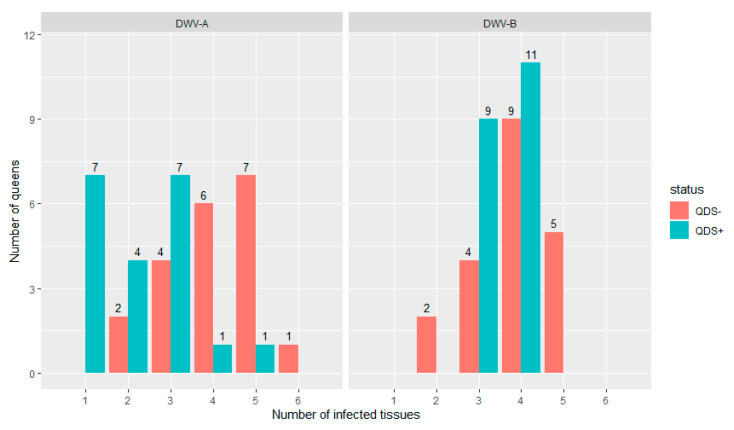
Distribution of the number of infected tissues in each queen for both DWV-A and DWV-B. Data is presented for queens descending from SOV-positive (QDS+, marked in green) and SOV-negative (QDS–, marked in red) queens. Queens descending from SOV+ colonies have a significantly lower number of infected tissues for DWV-A.

**Figure 2 viruses-13-01074-f002:**
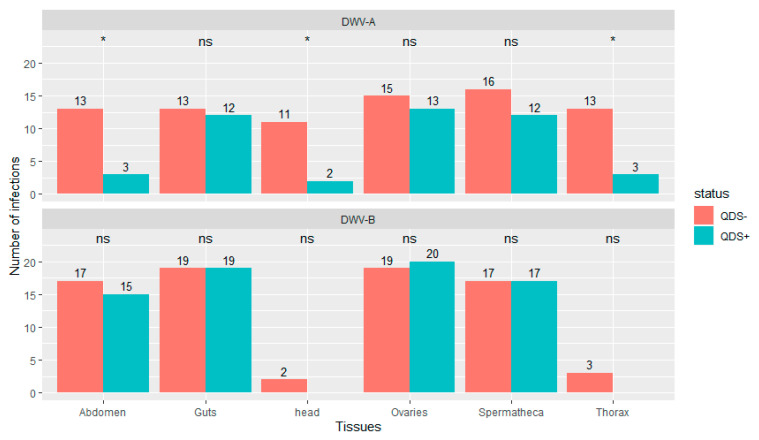
Number of tissues infected with DWV-A or DWV-B. Data is presented for QDS+ (green) and QDS– (red). Each group consists of 20 dissected queens. Significant differences between both groups are indicated at the top of each tissue with *, non-significant differences with ns.

**Figure 3 viruses-13-01074-f003:**
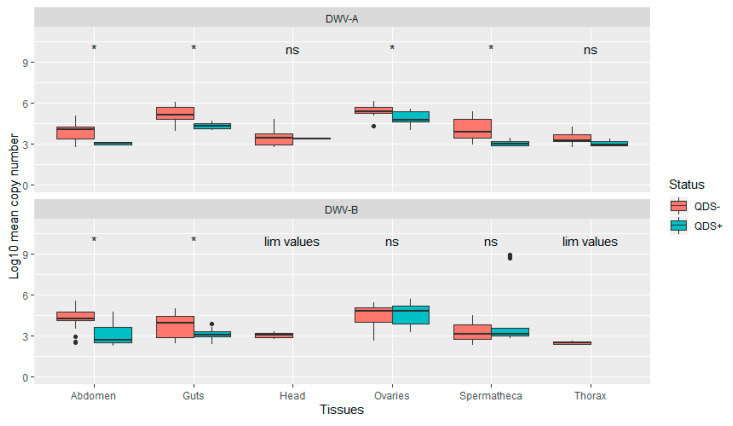
Boxplot of the infection load per tissue for DWV-A and DWV-B. Data are presented as Log10 copy number per sample for QDS+ (green) and QDS– (red). Significant differences between both groups are indicated at the top of each tissue with *, non-significant differences with ns. For the head and thorax insufficient samples were infected with DWV-B for statistical analyses (lim values).

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
