# Peer review of "Breeding for Virus Resistance and Its Effects on Deformed Wing Virus Infection Patterns in Honey Bee Queens"

_viruses, 2021, doi:10.3390/v13061074_

Round 1

Reviewer 1 Report

The manuscript is well written, and give novel information about SOV trait in the DWV quasispecies (A and B). 

The developed methodology is apropiate for the propose objetive .

The results are clear

The discussion gives valuable information about SOV trait and their relation with DWV quasispecies.

IMPORTANT:  revise the references numbers through the text, because some of them no coincidence with the numbers in references, such as 17, 18, 19, 27, 33 and 34. 
Revise all references.

Author Response

Reviewer 1:

General comment:

The manuscript is well written, and give novel information about SOV trait in the DWV quasispecies (A and B). The developed methodology is appropriate for the propose objective . The results are clear. The discussion gives valuable information about SOV trait and their relation with DWV quasispecies.

Specific comments:

Revise the references numbers through the text, because some of them no coincidence with the numbers in references, such as 17, 18, 19, 27, 33 and 34. Revise all references.

All references were thoroughly checked and adjusted were needed.

Reviewer 2 Report

General comments

The introduction is well written and helps to clarify important concepts in this area. The results and conclusions obtained in this study are novel are interesting.

Specific Comments

Lines 98-99

“All colonies were healthy, did not show signs of disease and were either free of viruses or infected with DWV only’

R: Since the levels of DWV change during the year, I think that it could be convenient clarify when were determined the viruses levels in these colonies.

Lines 193-194.

“In contrast to what was expected, no significant differences were found between QDS+ and QDS- in the reproductive tissues (ovaries and spermatheca), albeit being lower in QDS+ for DWV-A”

R: I am not sure if I understand this sentence. I understand that there are significant differences in reproductive tissues between DWV-A DWV-B.  Why the authors state that there are no significant differences? Please clarify.

Lines 222-224.

“The stronger effect of DWV-A compared to DWV-B might be attributed to the association between DWV-B and transmission via the varroa mite [17,19], reducing the importance of vertical transmission for the DWV-B genotype”.

R: This is an interesting proposal!

Lines 226-228.

“Possible infections occurring during larval feeding might be responsible for the high number of infected queens. Although larvae have a marked resistance to viral infection via oral transmission [17], transmission of viruses through larval feeding has been described [21]”.

R: I was unable to find direct evidence about larval transmission of DWV in the reference provided (Amiri et al., 2018) or in the references in that paper.  For example, although Yue et al. 2005, showed that DWV is present in larval food, they not showed actual transmission to the larvae.  I suggest Ryabov et al. “PeerJ vol. 4 e1591. 18 Jan. 2016, doi:10.7717/peerj.1591.

Author Response

Reviewer 2:

General comment:

The introduction is well written and helps to clarify important concepts in this area. The results and conclusions obtained in this study are novel are interesting.

Specific comments:

Lines 98-99: “All colonies were healthy, did not show signs of disease and were either free of viruses or infected with DWV only’

R: Since the levels of DWV change during the year, I think that it could be convenient clarify when were determined the viruses levels in these colonies.

Added ‘during the spring drone laying season according to’ in L102.

Lines 193-194: “In contrast to what was expected, no significant differences were found between QDS+ and QDS- in the reproductive tissues (ovaries and spermatheca), albeit being lower in QDS+ for DWV-A”

R: I am not sure if I understand this sentence. I understand that there are significant differences in reproductive tissues between DWV-A DWV-B.  Why the authors state that there are no significant differences? Please clarify.

L200 was adjusted to ‘In contrast to what was expected, no significant differences were found in the number of infected reproductive tissues (ovaries and spermatheca) between QDS+ and QDS- , albeit being lower in QDS+ for DWV-A’.

Lines 222-224: “The stronger effect of DWV-A compared to DWV-B might be attributed to the association between DWV-B and transmission via the varroa mite [17,19], reducing the importance of vertical transmission for the DWV-B genotype”.

R: This is an interesting proposal!

Thank you.

Lines 226-228: “Possible infections occurring during larval feeding might be responsible for the high number of infected queens. Although larvae have a marked resistance to viral infection via oral transmission [17], transmission of viruses through larval feeding has been described [21]”.

R: I was unable to find direct evidence about larval transmission of DWV in the reference provided (Amiri et al., 2018) or in the references in that paper.  For example, although Yue et al. 2005, showed that DWV is present in larval food, they not showed actual transmission to the larvae.  I suggest Ryabov et al. “PeerJ vol. 4 e1591. 18 Jan. 2016, doi:10.7717/peerj.1591.

Reference was added, original reference number was wrong due to a shift in the numbering.